# An Early SARS-CoV-2 Omicron Outbreak in a Dormitory in Saint Petersburg, Russia

**DOI:** 10.3390/v15071415

**Published:** 2023-06-22

**Authors:** Galya V. Klink, Daria Danilenko, Andrey B. Komissarov, Nikita Yolshin, Olga Shneider, Sergey Shcherbak, Elena Nabieva, Nikita Shvyrev, Nadezhda Konovalova, Alyona Zheltukhina, Artem Fadeev, Kseniya Komissarova, Andrey Ksenafontov, Tamila Musaeva, Veronika Eder, Maria Pisareva, Petr Nekrasov, Vladimir Shchur, Georgii A. Bazykin, Dmitry Lioznov

**Affiliations:** 1A.A. Kharkevich Institute for Information Transmission Problems of the Russian Academy of Sciences, 127051 Moscow, Russia; 2Smorodintsev Research Institute of Influenza, 197376 Saint-Petersburg, Russia; 3City Hospital #40, 197706 Saint-Petersburg, Russia; 4International Laboratory of Statistical and Computational Genomics, HSE University, 101000 Moscow, Russia; 5Skolkovo Institute of Science and Technology (Skoltech), 121205 Moscow, Russia; 6First Pavlov State Medical University, 197022 Saint-Petersburg, Russia

**Keywords:** SARS-CoV-2, Russia, founder effect, BA.1.1, superspreading, outbreak, public facility

## Abstract

The Omicron variant of SARS-CoV-2 rapidly spread worldwide in late 2021–early 2022, displacing the previously prevalent Delta variant. Before 16 December 2021, community transmission had already been observed in tens of countries globally. However, in Russia, the majority of reported cases at that time had been sporadic and associated with travel. Here, we report an Omicron outbreak at a student dormitory in Saint Petersburg between 16–29 December 2021, which was the earliest known instance of a large-scale community transmission in Russia. Out of the 465 sampled residents of the dormitory, 180 (38.7%) tested PCR-positive. Among the 118 residents for whom the variant had been tested by whole-genome sequencing, 111 (94.1%) were found to carry the Omicron variant. Among these 111 residents, 60 (54.1%) were vaccinated or had reported a previous infection of COVID-19. Phylogenetic analysis confirmed that the outbreak was caused by a single introduction of the BA.1.1 sub-lineage of the Omicron variant. The dormitory-derived clade constituted a significant proportion of BA.1.1 samples in Saint Petersburg and has spread to other regions of Russia and even to other countries. The rapid spread of the Omicron variant in a population with preexisting immunity to previous variants underlines its propensity for immune evasion.

## 1. Introduction

The Omicron variant of SARS-CoV-2 was first reported in South Africa on 24 November 2021 [1,2], and has been observed to rapidly spread worldwide soon thereafter. By mid-December, it outpaced the preceding diversity (mostly constituting the Delta variant) in many countries, including South Africa, the United Kingdom, Australia, and Canada, and became the prevalent variant [3]. While the SARS-CoV-2 epidemic in Russia between May and December 2021 was dominated by the Delta variant, with one particular Delta lineage, AY.122, having over 90% prevalence [4], by the end of January 2022, Omicron became the dominant variant in Russia (https://www.interfax.ru/russia/818539 (accessed on 1 June 2023)), leading to a rapid increase in morbidity and mortality (Appendix A). The details of its onset in Russia have been poorly studied. Hence, to repair this gap, we report an outbreak of the Omicron variant of COVID-19 in a student dormitory which occurred during the early weeks of the Omicron wave in Russia.

Among the nineteen full-genome Omicron samples obtained in Russia and deposited to the GISAID by 1 September 2022 with sampling dates between 3–15 December 2021, twelve were obtained from people with a known history of travel: 10 to the Republic of South Africa (all sampled on 3 December in Moscow), one to the Dominican Republic (sampled on 13 December in Saint Petersburg), and one to the Republic of the Congo (sampled on 10 December in Rostov-on-Don). Among the seven early genomic samples without a known travel history, six were not found to be associated with any other Russian sequences when placed on the UShER phylogenetic tree using the online UShER tool [5], i.e., representing Russian singletons [6]; while the seventh sequence was phylogenetically adjacent to the Rostov-on-Don sample with a travel history to the Republic of the Congo. Therefore, community transmission of the Omicron variant, if present, was at a low-level on those dates. Three of the nineteen samples belonged to the BA.1.1 lineage, including that obtained from the traveler to the Republic of the Congo.

While the Delta epidemic continued in Saint Petersburg throughout late 2021, with about 40,000 daily reported cases in November https://xn--80aesfpebagmfblc0a.xn--p1ai/ai/doc/1174/attach/2021-12-04_coronavirus_government_report.pdf (accessed on 1 June 2023), in Russian), we began systematic screening for early Omicron detection using the Ins214EPE assay [7] on the general population samples obtained from multiple hospitals and outpatient clinics [8] (Appendix A). Between 29 November‒15 December, we screened 200 to 1000 samples daily. Here, we describe an early outbreak of the BA.1.1 sub-lineage of Omicron in a student dormitory and investigate the spread of the virus inside the building and beyond.

## 2. Methods

The PCR screening procedure is described in [8].

We performed whole-genome sequencing (WGS) using the SARS-CoV-2 ARTIC V4 protocol and the Oxford Nanopore gridION (Oxford Nanopore Technologies plc.: Gosling Building, Edmund Halley Road, Oxford Science Park, OX4 4DQ, UK) or Illumina NextSeq 2000 (Illumina, Inc. Worldwide Headquarters: 5200 Illumina Way, San Diego, CA 92122 USA) sequencing technology. Consensus genome assembly was performed using bwa-mem and bcftools, and was preceded by adapter and primer trimming using trimmomatic, ivar (Illumina), andBAMClipper (Oxford Nanopore), along with custom scripts. An alternative allele was called if its read frequency exceeded 0.5 at a specific position. Positions with a coverage below 10 (Illumina) or 20 (Oxford Nanopore) were masked as N.

A few dormitory sequences assigned to the Omicron lineage were found to have positions that were termed as ancestral and/or Delta nucleotides. While these could be legitimately new mutations (including reversions), a close analysis of the NGS data hinted at the possibility that these could be artifacts of primer integration into the reads (leading to the reference variant), or contamination (or co-infection) with the Delta variant. To be on the conservative side, we marked such positions as N in the dormitory sequences for the purposes of tree construction.

For phylogenetic analysis, we downloaded the UShER SARS-CoV2 phylogenetic tree on 26 May 2022 and extracted a subtree of 572,763 BA.1.1 samples available in the GISAID. We then removed all dormitory samples from the tree and added their improved consensus to this tree with the UShER tool [5] and visualized it with iTOL [9]. The same was performed with seven dormitory samples of the Delta variant and the Delta subtree. Statistical analysis was performed with R [10]. Wilson confidence intervals were calculated with the Hmisc package [11], and plots were made using the tidyverse [12] and ggsignif [13] packages for R.

Custom scripts used in this work were deposited on GitHub (https://github.com/GalkaKlink/Omicron-Outbreak-in-Dorm (accessed on 1 June 2023)).

The number of daily new cases according to the Johns Hopkins University (JHU) Center for Systems Science and Engineering (CSSE) for Appendix A was obtained from https://github.com/CSSEGISandData/COVID-19_Unified-Dataset (accessed on 1 June 2023) [14].

## 3. Results

On 16 December 2021, in the course of screening, we detected Omicron in a hospital sample from a patient without a travel history. Follow-up contact tracing revealed that this sample came from a SARS-CoV-2 outbreak in a student dormitory in Saint Petersburg. This dormitory was located 500–700 m away from the university campus. According to the internal university regulations, since 1 September 2021, a hybrid mode of learning was implemented (consisting of online lectures and in-person practical courses). On 20 December 2021, all in-person activities were cancelled, and students were banned from leaving the dormitory. Between 17‒29 December 2021, we conducted follow-up testing of the dormitory residents. Out of the 465 residents, 180 (38.7%) tested positive for COVID-19 over these dates. For 137 samples, the Ins214EPE assay indicated that they were of the Omicron variant.

We performed whole-genome sequencing (WGS) for 118 samples with sufficiently low ct values. A total of 111 of the 118 sequences (94.1%) were classified as the Omicron variant on the basis of WGS. The remaining seven sequences were classified as non-Omicron (Delta variant).

### 3.1. Phylogenetic Distribution of Samples Indicates a Single Introduction into the Dormitory

All seven Delta samples were found to belong to AY.122, the predominant lineage observed in Saint Petersburg. Four of them formed a compact clade (transmission lineage [6]), while the remaining three were phylogenetically distinct (singletons [6]) (Appendix A). The fact that the Delta samples were scattered across the phylogeny of AY.122 is consistent with the multiple distinct sources of non-Omicron infection, in line with a high prevalence of the Delta variant in Saint Petersburg on those dates.

In contrast, all 111 Omicron samples were found to belong to the BA.1.1 sub-lineage and had a compact phylogenetic distribution within this sub-lineage (Figure 1). This is consistent with a single introduction and subsequent spread within the dormitory, or multiple infections from a single source.

The BA.1.1 sub-lineage is characterized by the R346K mutation on the spike protein (S:R346K). S:346 is an important immunogenic residue, and various mutations of it allow the virus to escape neutralization by multiple antibodies [15]. This site was shown to experience positive selection within the BA.1 sub-lineage of Omicron [16]. However, the arginine-to-lysine change observed in BA.1.1 is chemically conservative, does not lead to a major shift in antibody recognition, and does not confer a significant transmission advantage [17]. Therefore, the fact that the dormitory outbreak has been caused by the BA.1.1 sub-lineage rather than the ancestral BA.1 lineage is probably due to a founder effect. In any case, the extensive spread of a single Omicron sub-lineage, but with none of the three Delta sub-lineages is consistent with a higher transmission rate of the Omicron variant compared to the Delta variant in this setting.

All but five dormitory samples formed a single compact clade within the BA.1.1 sub-lineage, which we refer to as clade A (Figure 1). This clade was characterized with the C5812T synonymous mutation (nsp3:D1031D). Notably, the remaining five dormitory samples were positioned at the root of clade A, i.e., carrying all mutations of clade A except for the C5812T mutation; however, even for these samples, position 5812 was polymorphic, with the derived variant T present in 7–50% of the sequencing reads, suggesting that the C5812T mutation arose in the dormitory at the beginning of the outbreak.

We characterized the introduction and transmission of the virus in the dormitory outbreak using a phylodynamic approach. For this, we applied the birth-death skyline model [18] of BEAST2 [19] to the dormitory samples of Omicron. We considered three different fixed values of the clock rate: 0.75 × 10^−3^, 0.95 × 10^−3^, and 1.15 × 10^−3^, respectively [1,20]. In each scenario, the effective reproductive number R_e_ was estimated for three time periods: R_1_ before the first sample was collected on 16 December, R_2_ between 16 December and 24 December, and R_3_ between 24 December and 29 December 2021 (the date of the last sample collected).

The most recent common ancestor of the dormitory outbreak was estimated to be on 2 December with the 95% CI [23 November, 9 December] for the lowest value of the clock rate (0.75 × 10^−3^), 5 December [28 November, 11 December] for the intermediate value (0.95 × 10^−3^), and 7 December [1 December, 12 December] for the highest value of 1.15 × 10^−3^, respectively (Figure 2, Appendix A). Assuming a single introduction into the dormitory, which is supported by the monophyly of the dormitory samples, this implies that the infection was introduced into the dormitory about two weeks prior to the collection of the first sample on 16 December. The initial effective reproductive number R_1_ was found to be high across all three scenarios: 3.90 with 95% CI of [2.22, 5.67], 4.59 [2.56, 6.96], and 5.23 [2.82, 8.00] for the different clock rate values, respectively. Later, it was found to consistently drop by a factor of approximately 2.5 in all runs, with R_2_ being equal to 1.69 [1.00, 2.41], 1.83 [1.11, 2.58] and 1.97 [1.21, 2.74] respectively. R_3_ has a very wide credible interval which includes R_e_ = 1 and it is not informative regarding the phylodynamics between 25 December and 29 December. This was attributed to the low number of samples obtained from this time period.

The substantial fraction of infected individuals observed in the dormitory outbreak likely had some pre-existing immunity to SARS-CoV-2. Among the 137 patients who tested positive for Omicron, 71 (51.8%) reported previous infection or vaccination. This is in line with the high immune evasion properties of the Omicron variant [21,22].

### 3.2. An Elevated Risk of Within-Room Transmission

The corridor-type dormitory occupied a single nine-story building with natural ventilation. Most dormitory rooms had a four-bed layout, with up to twenty-four rooms per floor. Vent channels from the shared bathrooms joined the main air duct on the next floor. The total capacity of the dormitory was around 800 beds.

For 104 of the 111 Omicron-positive residents, the floor and room were both known. We assessed how the risk of transmission was affected by living together in the same room, or on the same floor with an infected individual. We reasoned that if the Omicron variant had been introduced into the dormitory just once, all differences between the samples originated during within-dormitory transmission. Therefore, the samples separated by a direct transmission of the virus are likely to have fewer differences compared to the samples separated by a chain of more than one transmission through other individuals.

To assess this, we calculated the mean pairwise phylogenetic distance *m* (which typically equals the number of single-nucleotide differences) between samples obtained from individuals residing in the same room or on the same floor and compared it with the expected distance between samples from the same floor or from anywhere in the building. To obtain these expected values, we reshuffled room labels across the individua 10,000 times while controlling and not controlling for the floor labels, and also reshuffled the floor labels separately.

We found that when two infected individuals resided in the same room, the phylogenetic distance between their SARS-CoV-2 samples was 1.8 times lower compared to an average pair of infected individuals residing on any floor (0.65 vs. 1.19, respectively, Figure 3A), and 1.5 times lower compared to individuals residing on the same floor (0.65 vs. 0.99, respectively, Figure 3B), and these differences were found to be significant (*p* = 0.0001 and *p* = 0.006, respectively). Conversely, accommodation on the same floor irrespective of the room did not lower the phylogenetic distance between the samples compared to pairs of infected individuals from anywhere in the building (1.18 vs. 1.18, respectively, *p* = 0.491, Figure 3C). These results indicate that residing in the same room with an infected individual increased the risk of transmission from that individual, while living on the same floor but in a different room exhibited no effects.

### 3.3. The Role of the Dormitory Outbreak in the Russian and Global Epidemic of Omicron

The BA.1.1 lineage comprised a considerable fraction of Russian samples in the beginning of 2022 (Figure 4A). The UShER tree of the BA.1.1 sub-lineage contained 489 non-dormitory Russian samples that were obtained after 16 December 2021. Among them, 51 (10%; Wilson 95% CI = 8–13%) belonged to clade A and carried all three of its characteristic mutations (Figure 1, Appendix A). Among the dormitory samples of clade A, 47.2% (50/106) were basal, i.e., carried no extra changes on top of the characteristic C5812T mutation of clade A. In contrast, all 51 non-dormitory samples carried extra mutations, indicating that they were exported from the dormitory into the general population of Saint Petersburg and beyond. According to the phylogenetic tree, there were at least three such exports of clade A (Figure 1). Clade A samples were most frequent in Saint Petersburg (comprising 18.6% of all Omicron samples in February 2022) as well as the surrounding Leningrad Oblast (Figure 4B and Figure 5), pointing to a considerable contribution of the dormitory outbreak to the Omicron wave in these areas. Meanwhile, the role of the dormitory outbreak in the spread of Omicron across most of the other regions of Russia regions was found to be negligible: for example, none of the 81 samples obtained from Moscow belonged to clade A (Figure 5).

In addition to the samples that originated from Russia, clade A also carried 118 non-Russian samples from 20 countries (Appendix A). All of them were collected after 16 December 2021. The fraction of such samples among all BA.1.1 samples was low (<<1%) in all countries except for Estonia, where it reached 1.8%. Notably, the two countries with the highest fraction, Estonia (1.8%; Wilson 95% CI = 0.8–3.9%) and Finland (0.5%; Wilson 95% CI = 0.3–0.8%), are geographically close to Saint Petersburg, and are frequent travel destinations for Saint Petersburg residents.

## 4. Discussion

In this work, we describe an outbreak of BA.1.1 in a student dormitory in Saint Petersburg at the beginning of the wave caused by the Omicron variant of COVID-19. Our study has some limitations that are standard for this kind of work. First, we relied on sequencing data and a phylogenetic tree, and therefore, mistakes in variant calling and tree reconstruction could have potentially affected our results. Second, without any information regarding the direct transmission of the virus between people, we were only able to judge the possibility of direct transmission between two people using the phylogenetic distances between their viral samples, which is not a very reliable method.

We show that the dormitory-derived variant spilled over into the general population of Saint Petersburg, representing a substantial fraction among the BA.1.1 samples that have been obtained from here. Additionally, it was found to have spread to several other regions of Russia and to other countries as well. As clade A differs from the root of BA.1.1 in three nucleotide mutations, all of which are synonymous, it is therefore unlikely that it had a fitness difference, meaning that most likely it has spread due to chance. The transmission of SARS-CoV2 is highly non-uniform, providing an important role for superspreading events in the epidemic [23,24]. Recently, using early Omicron transmission chains in Hong Kong, it was estimated that 80% of transmissions were generated by just 20% of cases, and the superspreading potential of Omicron was suggested to be higher than for the variants which circulated in 2020 [25]. In Russia, the whole Delta wave was mostly made by a single clade that has since likely spread due to chance [4].

Hotels and dormitories provide a major potential for superspreading. For Omicron, it was shown that even in a quarantine hotel, the virus moved between neighboring rooms through circulating air [26]. Nevertheless, in this study, we show that even in a student dormitory, where residents of different rooms are likely to actively communicate with each other, infection from roommates were found to be more likely than from other residents of the same floor or the entire building. It was previously shown that inside a hospital, health care workers operating in locations with a higher level of virus circulation are at greater risk of contracting the infection [27]. Therefore, living places with layouts carrying more beds per room may host more rapidly growing outbreaks than places with smaller rooms.

At the onset of Omicron in South Africa, its per day growth advantage over Delta was estimated to be 0.24 [1]. In agreement with this, while we detected four independent introductions of Delta in the dormitory simultaneously with the introduction of Omicron, though neither of these four introductions led to an outbreak. The higher transmissibility of Omicron is thought to be mainly due to its immune evasion properties [21,22]. In our study, near half of the dormitory residents with the Omicron variant were known to have been previously vaccinated or infected, thereby illustrating its potential for spread in a population with a pre-existing partial immunity.

Notably, despite the high fraction of dormitory-derived clade A in Saint Petersburg and Leningrad Oblast, its prevalence in other regions of Russia was low (Figure 5), pointing out that even close regions with a high passenger flow between them, such as Saint Petersburg and Moscow, have unique epidemiological histories of equally fit, viral variants. This is of course conditional depending on a similar fitness level of these variants; a novel advantageous variant can rapidly spread across a country and the world, as repeatedly observed during the SARS-CoV-2 pandemic.

## 5. Conclusions

We describe an early Omicron outbreak in a student dormitory in Saint Petersburg and reveal its substantial contribution to the early Omicron wave in Saint Petersburg, but not in Moscow, despite intensive traffic between these two cities. We also show that inside the building, an individual has a significantly greater chance of getting infected if they share a room with an infected person, despite the fact that students from different rooms are likely to actively communicate with each other.

## Figures and Tables

**Figure 1 viruses-15-01415-f001:**
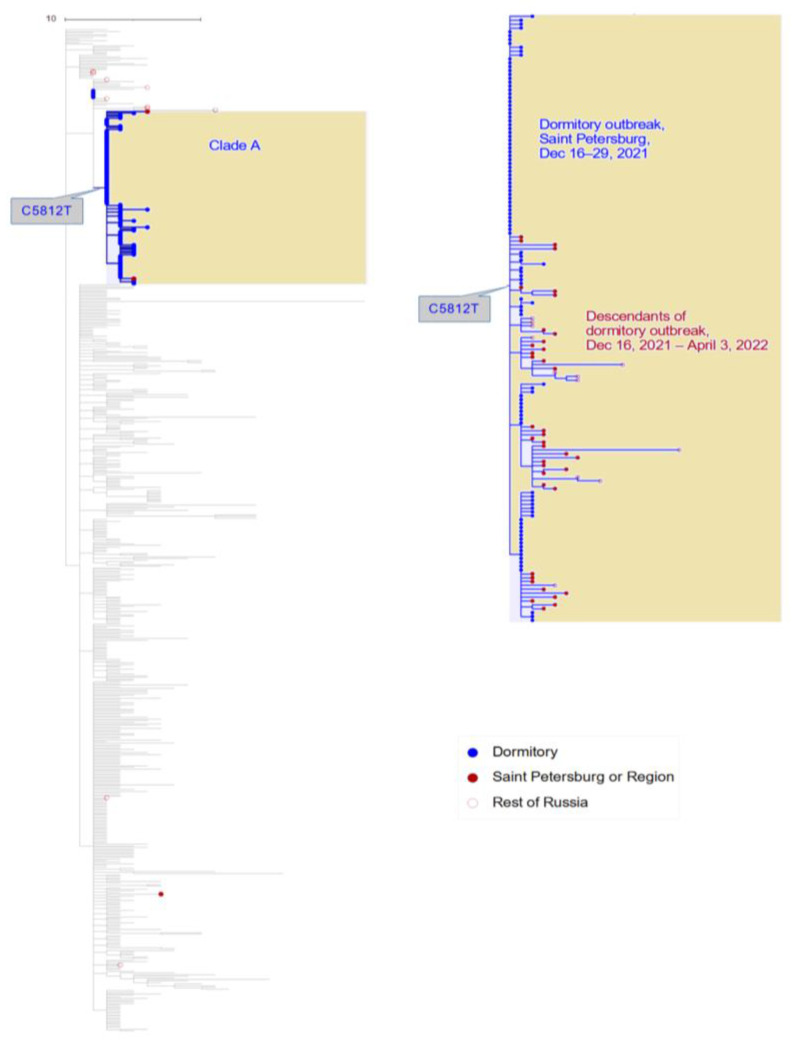
The Russian samples obtained between 3‒30 December 2021 and the Saint Petersburg dormitory outbreak on the global tree of BA.1.1. All 13 non-dormitory GISAID samples from Russia are shown (in red), together with a random sample of 400 (out of 8396) GISAID Omicron samples obtained in other countries (depicted in grey). Clade A, which is defined by the presence of the C5812T mutation, is shown in blue colour with yellow background. A zoom in of clade A including the outbreak samples (blue) is shown on the right, together with 51 descendant non-dormitory samples from Russia (red). Branch lengths are measured in number of mutations.

**Figure 2 viruses-15-01415-f002:**
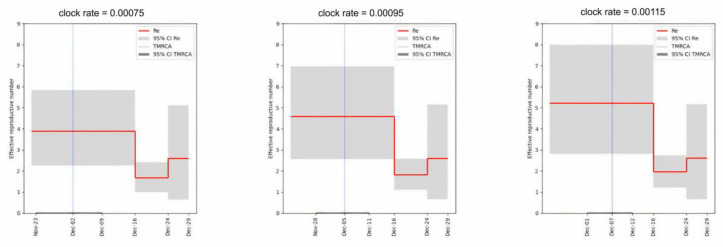
Skyline plots for the effective reproductive number R_e_ for different values of the molecular clock rate.

**Figure 3 viruses-15-01415-f003:**
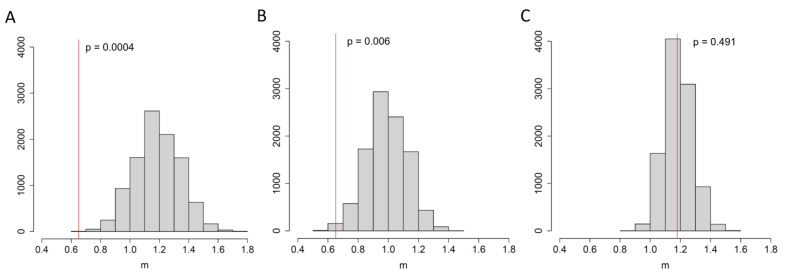
The mean phylogenetic distance *m* between two samples obtained from the same room (**A**,**B**) or floor (**C**) (red), compared to the expected distributions obtained by reshuffling the room labels independent of the floor (**A**), within the floor (**B**), or by reshuffling yhr floor labels (**C**) of the samples. P represents the fraction of reshuffling trials with a mean phylogenetic distance below *m*. The Y-axis represents the number of reshufflings.

**Figure 4 viruses-15-01415-f004:**
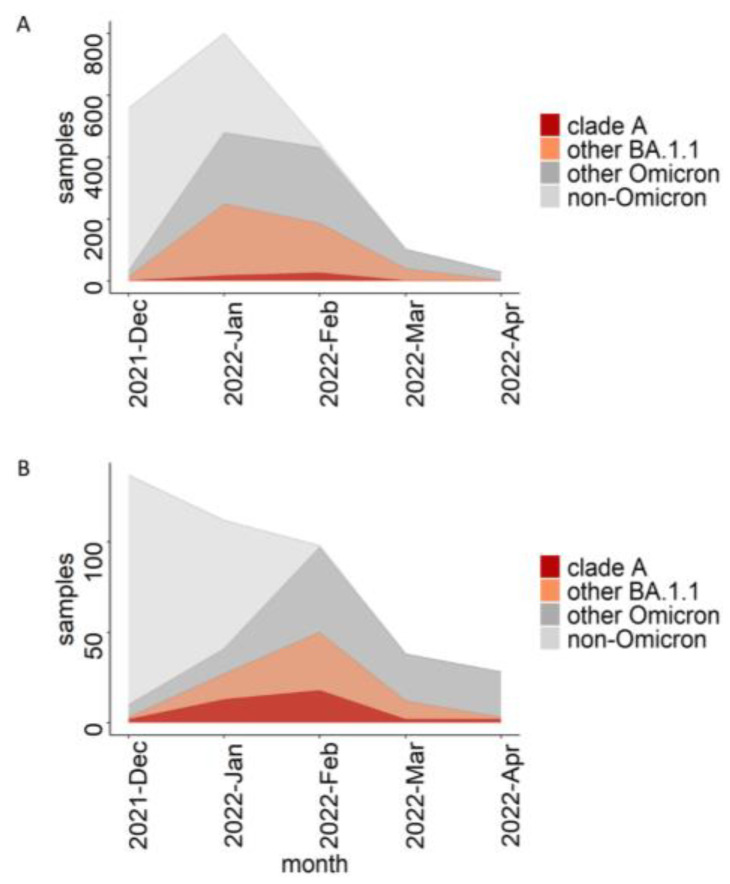
The fraction of clade A, BA.1.1, and Omicron samples among the Russian (**A**) and Saint Petersburg (**B**) samples obtained from the GISAID included in the UShER phylogenetic tree downloaded on 26 May 2022. Samples from the dormitory are not included.

**Figure 5 viruses-15-01415-f005:**
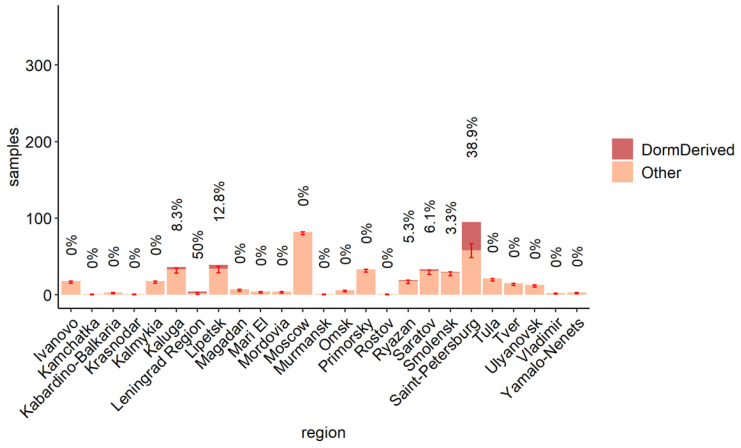
Clade A samples (orange) among all BA.1.1 samples across all regions of Russia. Numbers on bars are the percentage of clade A samples in each region; 95% Wilson CIs are shown as bars. Dormitory samples are not included.

## Data Availability

The datasets generated and analyzed in the current study are available in the GISAID.

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
