# Peer review of "An Early SARS-CoV-2 Omicron Outbreak in a Dormitory in Saint Petersburg, Russia"

_viruses, 2023, doi:10.3390/v15071415_

Round 1

Reviewer 1 Report

GENERAL FEED-BACK

This study investigated a COVID-19 outbreak in a dormitory in Saint Petersburg during 17-27 December 2021, when Delta variant was still predominant in Russia, by contact tracing following a hospital admission of a COVID-19 patient on 16 Dec 2021 infected by the Omicron variant.

Out of 465 dormitory residents, 180 tested positive for COVID-19, 137 of which were omicron cases. Whole-genome sequencing on a sub-set of 118 samples 1 with low enough CT yielded 111 cases classified as Omicron.

All 111 dormitory omicron samples were sub-variant BA.1.1 referring to a single clade (clade A), probably originating from a single introduction of a BA.1.1 case in the dorm.

54.1% of 111 omicron residents had humoral immunity from vaccination or previous SARS-CoV-2 infection.

Phylogenetic analysis showed that residing in the same dormitory room with an infected individual increased the risk of transmission from that individual, whereas living on the same floor but in a different room had no effect.

The dormitory outbreak considerably contributed to the Omicron wave in the city.

The study is interesting, as it provides epidemiological figures on a outbreak from a large European city. By contrast, the impact of the dormitory outbreak to the rest of the country was negligible.

SOME COMMENTS

·        Lines 33-41: here it is worth mentioning that Omicron not only displaced previous variants but also increased the risk of infections as well as reinfections in highly yet systematically screened population as health care workers, due to evasion of humoral immunity response, especially outside heath care premises (non-occupational infection) PMID: 36016284 PMID: 35215930

·        Figure 1: a different contrasting colour could be used to visualize Clade A, e.g. yellow

·        An epidemic curve would be useful to visualize the epidemic over time

·        What was the definition of “vaccinated” individuals? What type of vaccine and how many doses?

·        Some English revisions required: e.g. Figure 1 legend (“gray” should be “grey”)

Line 243/44; “its estimated per day growth advantage over 243 Delta was estimated to be 0.24” awkward expression

Etc.

Author Response

Thank you for your comments. Please, find our reply in attachment.

Reviewer 2 Report

Please address the following points:

Line 44, please mention the year for the December sampling dates.

In the Introduction section, it would be useful -if such data exists- to include a graph of the omicron detection over the 2021-2022, and highlight the window of time that the current investigation takes place.

In the introduction section, please provide a bit more description of the student dormitory (e.g., total capacity, frequency of travel for students, relative distances of travel, air conditioning system between rooms, etc) - alternatively you can add these details to section 3.2

In the introduction you mention sampling 200 - 1000 samples daily. That is absolutely ok, but the sampling strategy should be explained in the methodology, as it is important to understand both who was sampled and why - as well as who was not sampled and why.

In the methodology, are the custom scripts accessible somewhere? Have they been deposited?

In the methodology section, one presumes that the primer integration was identifiable by the primer sequence? Do not disagree with the conservative approach, just trying to understand how the call was made for primer integration specifically.

In the discussion section, are there any other published manuscripts on the spread of Omicron variant within dormitories, that you can reference? It would allow to understand how your work (which is unique to Russia) compares to other parts of the world.

After line 257 please add 3-4 sentences on the limitations of this study.

After the limitations sentences, please add a conclusion section. The discussion is well done, but the manuscript still needs a conclusion summarising the entire work.

Author Response

(The authors gave the same response as above.)
